# Effect of Body Fat Percentage on Muscle Damage Induced by High-Intensity Eccentric Exercise

**DOI:** 10.3390/ijerph17103476

**Published:** 2020-05-16

**Authors:** Eun-Jung Yoon, Jooyoung Kim

**Affiliations:** 1Department of Physical Education, Korea National University of Education, Cheongju-si 28173, Korea; 315836@naver.com; 2Office of Academic Affairs, Konkuk University, Chungju-si 27478, Korea

**Keywords:** body fat percentage, creatine kinase, eccentric exercise, muscle damage, myoglobin

## Abstract

This study aimed to investigate the impact of percent body fat (%BF) on muscle damage after high-intensity eccentric exercise. Thirty healthy male undergraduates (mean age: 22.0 ± 2 years, height: 176.9 ± 5 cm, weight: 75.8 ± 11.6 kg) participated in this study, and they were classified according to their %BF into a high %fat group (HFG, ≥20%, *n* = 15) and a low %fat group (LFG, ≤15%, *n* = 15). For eccentric exercise, two sets of 25 reps were performed on a modified preacher curl machine using the elbow flexor muscle. Maximal isometric strength, muscle soreness (passive and active), creatine kinase (CK), and myoglobin (Mb) were measured as indices of muscle damage. The data were analyzed with repeated measures ANOVA. The results show that there is a significant group–time interaction for both CK and Mb after eccentric exercise (*p* = 0.007, *p* = 0.015, respectively), with a greater increase in the HFG than in the LFG. However, there was no significant group–time interaction for maximal isometric strength and muscle soreness (passive and active) (*p* > 0.05). These results suggest that %BF is a factor that alters the muscle damage indices CK and Mb, which indicate membrane disruption, after eccentric exercise.

## 1. Introduction

Repeated unaccustomed or high-intensity eccentric exercise is well known to induce muscle damage [1,2]. Muscle damage not only lowers maximal strength but also causes delayed onset muscle soreness (DOMS) and increases the release of muscle-specific proteins such as creatine kinase (CK) and myoglobin (Mb) into the bloodstream [3,4].

Some animal studies suggested that obesity is a potential factor that weakens the cell membrane by affecting membrane biomechanical properties [5,6]. Obesity is a state of excessive accumulation of adipose tissue within the body [7]. If excessive fat increases the saturation of fatty acyl chains in the sarcolemma, phospholipid packing density is also increased, leading to a rigid membrane [8]. Such changes weaken the cell membrane and elevate the susceptibility to damage [5]. Moreover, adipose tissues are the source of pro-inflammatory cytokines such as tumor necrosis factor-α (TNF-α), interleukin-1β (IL-1β), and interleukin-6 (IL-6), and these cytokines create a low-grade chronic inflammatory condition and induces oxidative stress [9,10]. Oxidative stress facilitates phagocyte infiltration of damaged muscle tissues, thereby causing further membrane disruption [11]. In fact, some human studies reported a significant positive correlation between neutrophil responses and increased IL-6, CK, and Mb levels after exercise [11,12,13].

Several human studies have investigated the impact of body composition on muscle damage following eccentric exercise [9,14,15,16]. Hickner et al. [14] reported that individuals with a high percentage of body fat (%BF) show more pronounced loss of leg strength after eccentric exercise-induced muscle damage. Paschalis et al. [16] reported that the overweight women group had decreased maximal strength and increased muscle soreness and CK compared to the normal weight women group. Some recent studies also found that the group with a high body mass index (BMI, ≥25 kg/m^2^) or %BF (≥30.1%) had greater increases in muscle damage parameters following eccentric exercise compared to the normal group [9,15]. Taken together, body composition is certainly a potential factor that affects the degree of muscle exercise following eccentric exercise.

However, Kim and So [15] and Paschalis et al. [16] used BMI as the parameter for body composition assessment. Although BMI is easy to measure, it leads to a less accurate body composition assessment because there are people who have a high BMI due to their high muscle mass and also people who have a normal BMI but have a high %BF [9,17]. The current BMI has several limitations, as it may not be appropriate for different races and does not accurately reflect adipose tissue [18]. According to a recent study, %BF enables a more accurate body assessment for overweight and obesity than does BMI [19]. Moreover, to the best of our knowledge, no study has investigated the relationship between %BF and muscle damage following eccentric exercise in the Korean population. 

Therefore, this study aimed to investigate the impact of %BF on muscle damage parameters such as maximal isometric strength, muscle soreness, CK, and Mb after high-intensity eccentric exercise using an elbow flexor model.

## 2. Materials and Methods

### 2.1. Subjects

Thirty healthy male undergraduates (mean age: 22.0 ± 2 years, height: 176.9 ± 5 cm, weight: 75.8 ± 11.6 kg) participated in this study. Sample size was computed with reference to a prior study [14], under a statistical power of 0.80, alpha error probability of 0.05, and effect size of 0.25 (G*power, version 3.0.10, Heinrich-Heine University, Dusseldorf, Germany). The participants had not engaged in regular resistance exercise for the past six months and currently had no neurological or musculoskeletal disorders. They were non-smokers and were not currently taking proteins or vitamin supplements. The participants were informed of the purpose and procedure of the study by the researcher, after which they voluntarily signed the informed consent form approved by the Institutional Review Board. The participants were instructed not to take anti-inflammatory drugs or analgesics and not to engage in high-intensity or unaccustomed physical activity or exercise during the experiment period. We divided the participants according to their %BF into the high %fat group (HFG, ≥20%, *n* = 15) and low %fat group (LFG, ≤15%, *n* = 15). The classification was performed with reference to the low and high %fat criteria presented by a past study on the Korean population [20]. The characteristics of each group are shown in Table 1. 

### 2.2. Eccentric Exercise

In this study, we used the elbow flexor model for eccentric exercise. The elbow flexor model has been widely used in muscle damage studies [21,22,23]. While sitting on the seat of a modified preacher curl machine (EMC model, Kookmin University, Seoul, Korea), the participants placed their non-dominant arm on the machine pad with an elbow joint angle of 90 degrees. Then, when the researcher pulled the lever attached to the pad, the participants were to resist the force and pull the pad toward their body with maximal force. Two sets of 25 reps of elbow flexor muscle contraction were performed. Each elbow flexor muscle contraction was maintained for three seconds, followed by a 12-s rest. To measure seconds, a stopwatch (KS-201, KORSPO, Namyangju, Korea) was used. A five-minute rest was taken between sets. We referred to previous studies [12,15] for all protocol related to eccentric exercise.

### 2.3. Body Composition

Body composition was measured using InBody 270 (InBody CO., LTD., Seoul, Korea), which utilizes the bioelectrical impedance analysis (BIA). BIA is an efficient tool for accurate assessment of %BF, the validity and reliability of which have been confirmed by previous studies [24,25]. The measurement was taken in the morning of the first day of the experiment following eight hours of fasting. The participants wore shorts and t-shirts, wiped their hands and feet with tissue, and stood on the foot plate of the device with both feet and arms open for the measurement. This posture was maintained until the test was finished. To ensure the accuracy of %fat measurement, the participants were instructed to refrain from vigorous exercise or physical activity as well as sauna or prolonged bathing on the day before the measurement.

### 2.4. Maximal Isometric Strength

The maximal isomeric strength of the elbow flexor was measured after attaching a strain gauge (PKS-1250, Poong Kwang, Seoul, Korea) on a modified preacher curl machine. We referred to a prior study [12] for the measurement method and procedure. As with eccentric exercise, we asked the participants about their non-dominant arm prior to measuring maximal isometric strength. After placing their arm on the machine pad with their elbow joint set at 90 degrees, they pulled the pad toward their body with maximal force. The value was recorded on the monitor of the strain gauge. Three measurements were taken, with a one-minute rest between measurements. The average of the three measurements was computed and converted to a relative value (%). Maximal isometric strength was measured before exercise and immediately, 24 h, 48 h, 72 h, and 96 h after exercise.

### 2.5. Muscle Soreness

The visual analogue scale (VAS) is the most classic instrument used to measure muscle soreness. A straight 100-mm line is drawn, with the left-most point (0 mm) indicating absolutely no soreness and the right-most point (100 mm) indicating the most severe soreness. VAS has been widely utilized in studies on exercise-induced muscle damage because it is easy to use in a variety of environments and can be applied with only little practice [26,27]. In this study, we measured muscle soreness with two methods (passive and active). First, passive muscle soreness was measured by having the participant vertically mark the level of their muscle soreness at rest on the VAS. Then, active muscle soreness was measured by having the participant perform two reps of curl with a 1-kg dumbbell with the arm that had performed the eccentric exercise and vertically marking the level of their muscle soreness on the VAS. Muscle soreness was measured before exercise, and 24 h, 48 h, 72 h, and 96 h after exercise.

### 2.6. CK and Mb

Blood samples were taken from the forearm vein before exercise, and 24 h, 48 h, 72 h, and 96 h after exercise to obtain blood CK and Mb levels. A vacutainer system was used for the sampling, and 5 mL of blood was taken for each sampling. The sample contained in a vacutainer SST was centrifuged at 2500–3000 rpm for 10–15 min. Then, the sample was tested for CK by placing the reagent (Vitros 750, Johnson and Johnson, Phoenix, AZ, USA) in an automated blood analyzer (Ortho Johnson Vitros DT60 II, Johnson and Johnson, Phoenix, AZ, USA). The sample was tested for Mb by placing the reagent (LIAISON Myoglobin, DiaSorin, Saluggia, Italy) in clinical test equipment (LIAISON, DiaSorin, Saluggia, Italy).

### 2.7. Statistical Analysis

Statistical analyses were performed using the IBM SPSS statistics software (SPSS ver. 21.0, IBM, Armonk, NY, USA). All results were presented as the mean and standard deviation. Time (pre, post, 24 h, 48 h, 72 h, 96 h) and group (HFG and LHG) interaction was analyzed with repeated-measures ANOVA, and when a significant interaction was found, an independent sample *t*-test was performed as a post-hoc test. Effect size (ES) was calculated using Cohen *d* with pooled standard deviation. Statistical significance was set at 0.05.

## 3. Results

### 3.1. Maximal Isometric Strength Following Eccentric Exercise According to %Fat

Table 2 shows the changes in maximal isometric strength following eccentric exercise according to %fat. Both groups had significant reductions in maximal isometric strength by more than 40% immediately after eccentric exercise (HFG: 55.4% reduced; LFG: 46.7% reduced). The main effect of time was significant (F = 62.582, *p* < 0.001), but the main effects of group (F = 0.221, *p* = 0.642) and time–group interaction (F = 0.847, *p* = 0.519) were not significant. 

### 3.2. Passive Muscle Soreness Following Eccentric Exercise According to %Fat

Table 3 shows the changes in passive muscle soreness following eccentric exercise according to %fat. Both groups showed a significant increase in passive muscle soreness after eccentric exercise, which peaked at 48 h after exercise followed by a gradual decline. The main effect of time was significant (F = 34.488, *p* < 0.001), but the main effects of group (F = 0.468, *p* = 0.499) and time–group interaction (F = 0.426, *p* = 0.790) were not significant. 

### 3.3. Active Muscle Soreness Following Eccentric Exercise According to %Fat

Table 4 shows the changes in active muscle soreness following eccentric exercise according to %fat. Active muscle soreness also peaked at 48 h after exercise followed by a gradual decline. The main effect of time was significant (F = 55.067, *p* < 0.001), but the main effects of group (F = 0.634, *p* = 0.432) and time–group interaction (F = 0.651, *p* = 0.627) were not significant.

### 3.4. CK after Eccentric Exercise According to %Fat

Table 5 shows the changes of CK after eccentric exercise according to %fat. CK continuously rose until 96 h after exercise in both groups. The main effect of time (F = 4.762, *p* < 0.001) as well as the main effects of group (F = 7.403, *p* = 0.011) and time–group interaction (F = 3.721, *p* = 0.007) were significant. The post-hoc tests showed that the HFG had higher CK levels than that of LFG at 24 (ES: 0.8; 95% confidence interval (CI): 160.0, 2690.9), 48 (ES: 0.9; 95% CI: 631.5, 5368.6), and 72 (ES: 0.7; 95% CI: 405.4, 23375.3) hours after exercise. 

### 3.5. Mb after Eccentric Exercise According to %Fat

Table 6 shows the changes of Mb after eccentric exercise according to %fat. Mb continuously rose until 96 h after exercise in both groups. The main effect of time (F = 4.511, *p* < 0.01), as well as the main effects of group (F = 4.728, *p* = 0.038) and time–group interaction (F = 3.233, *p* = 0.015) were significant. The post-hoc tests showed that the HFG had higher Mb levels than that of LFG at 24 (ES: 0.9; 95% CI: 31.1, 289.8) and 72 (ES: 0.7; 95% CI: 53.5, 1665.6) hours after exercise.

## 4. Discussion

This study aimed to investigate the impact of %BF on high-intensity eccentric exercise-induced muscle damage. The results show that the high %BF group (HFG) had higher CK and Mb levels during the recovery period following eccentric exercise compared to the low %BF group (LFG). These results show that %BF may have a potential impact on muscle damage following eccentric exercise. Some past studies reported similar results [9,15,16]. For instance, the elevation of CK after eccentric exercise was greater in overweight or obese people as determined by BMI [15,16]. A recent study [9] also found that women with a high %BF have higher CK levels after eccentric exercise compared to lean women. Based on the current obesity criteria (%BF: ≥25%, BMI: ≥25 kg/m^2^), we compared the mean %BF (25.1%) and BMI (26.0 kg/m^2^) in the high %fat group (HFG) and found that it satisfies the obesity criteria. The elevation of CK after eccentric exercise is closely related to membrane disruption. CK and Mb are blood parameters frequently utilized to indirectly examine membrane disruption after eccentric exercise-induced muscle damage, and their levels may be increased fourfold after exercise from the baseline levels [28]. In this study, after eccentric exercise in HFG, CK was shown to be 12848.6 U/L on average at 96 h after exercise, while Mb was 958.9 ng/mL on average at 72 h after exercise, showing high peak levels. This is similar to the CK and Mb levels shown in previous studies [29,30], suggesting that eccentric exercise caused severe muscle damage. Of the potential causes of CK and Mb increase following eccentric exercise, inflammatory response is the most well-known [9,15]. Several studies have reported that there is significant association between the post-eccentric-exercise increases in inflammatory cell activity and membrane disruption indicator [11,12]. 

Another study reported that there is a significant correlation between the elevation of oxidative stress marker malondialdehyde (MDA) and CK after high-intensity exercise [31]. A recent study [9] observed that individuals with body fat higher than 30.1% have escalated protein carbonyl levels after eccentric exercise. In addition, obesity may increase oxidative stress and hinder normal calcium signaling, and these changes are known to accelerate membrane disruption by facilitating calcium leakage from the sarcoplasmic reticulum [32,33]. If calcium is leaked and consequently is present in elevated levels intracellularly, the calcium homeostasis is disturbed, which in turn activates calcium-dependent cystein proteases such as calpain, ultimately facilitating skeletal muscle protein breakdown [34]. Based on the previously mentioned studies, we can infer that the potential cause of the escalated CK and Mb levels in the HFG in our study involves the inflammatory response and oxidative stress that ensued eccentric exercise. However, we did not directly examine this in this study, so further studies are needed to support this hypothesis. 

On the other hand, there were no significant differences in muscle soreness (both passive and active) and maximal isometric strength between the two groups. Muscle soreness is an indirect sign of inflammatory response following eccentric exercise [12], and maximal isometric strength is an important marker for muscle recovery [35]. Considering these features, their results should have been in line with the changes of CK and Mb levels, but completely different patterns were observed. In contrast to our findings, previous studies reported that the group with high BMI or %BF had high DOMS after eccentric exercise and had delayed recovery of maximal isometric strength [9,15,16]. Such inconsistency in the study findings may be attributable to a variety of factors, such as differences in body composition measurement, eccentric exercise model, and participants′ race. Regarding the inconsistency in muscle damage markers, some studies [36,37] reported that DOMS or maximal isometric strength may not adequately reflect other indices of muscle damage after eccentric exercise. This suggests that the presence of significant changes in CK or Mb levels indicating muscle damage after eccentric exercise does not necessarily mean that the same pattern of change is observed with DOMS or maximal isometric strength. Inconsistency with changes in the muscle damage indicator was also observed in a similar pattern by a few other studies. One study reported that maximum isometric strength returned to the baseline 4 days after a long-distance triathlon while CK continued to increase [38], and another study reported that even though direct damage occurred to muscle fibers in all groups after exercise-induced muscle damage, there was no significant difference in muscle soreness between the groups [39]. It is not clear what causes such inconsistency.

This study has a few limitations. First, we did not examine participants’ diets. Diet patterns including fat intake can also influence muscle damage [40]. If we had examined diet, we would have been able to interpret our findings from more diverse angles based on the information about participants’ fat intake frequency and percentage. Second, %BF was measured via BIA in this study. Although BIA is the general method for measuring %BF, it does have its limitations. Subsequent studies should investigate %BF using more accurate methods such as DEXA and CT scan. Third, muscle soreness was measured only by VAS in this study. In particular, passive muscle soreness was measured in a stable condition. As it is possible that muscle soreness may not be felt in a condition where exercised muscles do not engage, mechanical stimuli such as palpation, contraction or stretching which can induce muscle soreness should be provided to muscles [41]. In this study, a couple of studies suggested that it would be better to include VAS, category ratio-10 scale (CR-10), and pressure pain threshold (PPT) all together when evaluating muscle soreness after eccentric exercise [42,43]. A future study may need to use various pain measurement methods along with VAS. Finally, we did not measure any markers that could be used to test hypotheses pertinent to inflammatory responses, oxidative stress, or calcium in this study. Despite placing significance in these responses in our discussion of the study findings, we did not measure any of relevant markers, so they still remain as hypotheses. These limitations should be addressed in future studies. 

## 5. Conclusions

The findings of this study suggest that %BF is a potential factor that alters CK and Mb levels, which indicate membrane disruption and are parameters of muscle damage following eccentric exercise. Therefore, after a high %BF or obese individual performs eccentric exercise, a sufficient recovery time or effective intervention (e.g., physical modality or nutrition) must be provided until the next exercise session. Subsequent studies should clearly test various hypotheses pertinent to the relationship between %BF and eccentric exercise-induced muscle damage using more diverse parameters such as inflammatory responses and oxidative stress.

## Figures and Tables

**Table 1 ijerph-17-03476-t001:** Participant characteristics by group.

Variable	HFG (*n* = 15)	LFG (*n* = 15)	*p*
Age (yrs)	22.5 ± 2	21.4 ± 2	*p* > 0.05
Height (cm)	176.9 ± 5	176.6 ± 6	*p* > 0.05
Weight (kg)	81.7 ± 12.2	69.9 ± 7.6	*p* < 0.01 **
Body fat (%)	25.1 ± 3.5	12.1 ± 1.2	*p* < 0.001 ***
BMI (kg/m^2^)	26.0 ± 3.0	22.3 ± 1.8	*p* < 0.001 ***

Values are Mean ± SD; HFG, high %fat group; LFG, low %fat group, body mass index (BMI); ** *p* < 0.01, *** *p* < 0.001; tested by independent sample *t*-test.

**Table 2 ijerph-17-03476-t002:** Maximal isometric strength following eccentric exercise according to %fat.

Unit: %	Pre	Post	24 h	48 h	72 h	96 h	*p*
HFG (*n* = 15)	100 ± 0.0	44.6 ± 17.1	58.4 ± 23.3	62.6 ± 28.5	68.6 ± 25.1	80.4 ± 35.0	0.519
LFG (*n* = 15)	100 ± 0.0	53.3 ± 13.0	65.2 ± 17.8	62.6 ± 19.1	70.2 ± 18.1	77.8 ± 17.1	
Effect size (95% CI)	-	0.5 (−2.6, 20.0)	0.3 (−8.7, 22.3)	0.0 (−18.1, 18.1)	0.1 (−14.7, 17.9)	0.1 (−18.0, 23.2)	

Values are Mean ± SD; HFG, high %fat group; LFG, low %fat group; CI, confidence interval.

**Table 3 ijerph-17-03476-t003:** Passive muscle soreness following eccentric exercise according to %fat.

Unit: mm	Pre	24 h	48 h	72 h	96 h	*p*
HFG (*n* = 15)	0.0 ± 0.0	35.6 ± 21.6	44.6 ± 29.4	34.6 ± 27.1	24.2 ± 26.6	0.790
LFG (*n* = 15)	0.0 ± 0.0	28.1 ± 15.5	37.5 ± 15.8	31.3 ± 15.2	23.6 ± 20.6	
Effect size (95% CI)	-	0.3 (−6.5, 21.5)	0.3 (−10.5, 24.7)	0.1 (−13.1, 19.7)	0.0 (−17.1, 18.3)	

Values are Mean ± SD; HFG, high %fat group; LFG, low %fat group; CI, confidence interval.

**Table 4 ijerph-17-03476-t004:** Active muscle soreness following eccentric exercise according to %fat.

Unit: mm	Pre	24 h	48 h	72 h	96 h	*p*
HFG (*n* = 15)	0.0 ± 0.0	50.3 ± 19.3	56.0 ± 27.4	43.7 ± 28.2	26.2 ± 25.9	0.627
LFG (*n* = 15)	0.0 ± 0.0	40.6 ± 15.9	48.7 ± 15.7	39.8 ± 15.5	27.0 ± 21.5	
Effect size (95% CI)	-	0.5 (−3.5, 22.9)	0.3 (−9.4, 24.0)	0.1 (−13.1, 20.9)	0.0 (−17.0, 18.6)	

Values are Mean ± SD; HFG, high %fat group; LFG, low %fat group; CI, confidence interval.

**Table 5 ijerph-17-03476-t005:** CK after eccentric exercise according to %fat.

Unit: U/L	Pre	24 h	48 h	72 h	96 h	*p*
HFG (*n* = 15)	129.7 ± 38.2	1619.9 ± 2391.6 *	3469.5 ± 4389.9 *	7474.7 ± 9768.9 *	12,848.6 ± 21,608.3	0.007
LFG (*n* = 15)	130.9 ± 37.3	194.4 ± 66.9	469.4 ± 885.1	491.9 ± 775.7	958.2 ± 2149.3	
Effect size (95% CI)	-	0.8 (160.0, 2690.9)	0.9 (631.5, 5368.6)	1.0 (1799.8, 12,165.7)	0.7 (405.4, 23,375.3)	

Values are Mean ± SD; HFG, high %fat group; LFG, low %fat group; CI, confidence interval; * *p* < 0.05; tested by independent sample *t*-test.

**Table 6 ijerph-17-03476-t006:** Mb after eccentric exercise according to %fat.

Unit: ng/mL	Pre	24 h	48 h	72 h	96 h	*p*
HFG (*n* = 15)	30.0 ± 15.7	201.0 ± 242.7 *	662.7 ± 1137.5	958.9 ± 1511.4 *	472.1 ± 603.3	0.015
LFG (*n* = 15)	24.3 ± 7.3	40.5 ± 30.8	88.2 ± 210.0	99.3 ± 196.4	225.6 ± 524.4	
Effect size (95% CI)	-	0.9 (31.1, 289.8)	0.7 (−37.2, 1186.2)	0.7 (53.5, 1665.6)	0.4 (−176.2, 669.2)	

Values are Mean ± SD; HFG, high %fat group; LFG, low %fat group; CI, confidence interval; * *p* < 0.05; tested by independent sample *t*-test.

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
