# Peer review of "Effect of Body Fat Percentage on Muscle Damage Induced by High-Intensity Eccentric Exercise"

_ijerph, 2020, doi:10.3390/ijerph17103476_

Round 1
Reviewer 1 Report
The article has good relevance, however it needs some adjustments according to the attached document

Reviewer 2 Report
I would like to commend the authors to begin with for an interesting manuscript that I found very well presented and easy to read. However, I do have some concerns over the validity of the results due to the methodology employed that I will summarise below:
General comments:
- Greater care is required with using terminology interchangeably, eg. 'Muscle Injury' / DOMS / membrane damage. Be consistent. I would remove the term muscle injury as this is misleading, and refer to exercise induced muscle damage, of which membrane disruption, reduced strength, delayed onset soreness and increased CK/Mb are characteristics.
Specific Comments:
- Introduction:
- The introduction felt a little brief. Line 15-25 from the Discussion should be removed and put into the introduction to give greater explanation of the processes occurring and relevance of body fat to muscle membrane damage.
- Greater discussion of BMI vs BF% would have improved the section.
2. Materials and Methods:
- State the type of research design eg prospective cohort study
- No description of the validity and reliability of BF% as a measure.
- Eccentric protocol: was it truly eccentric? The description sounds more isometric. 25 maximal contractions sounds unlikely. This could have been worked out relative to an MVC for greater accuracy.
- Maximal isometric strength measurement: Picture of setup would benefit the reader. How long did they pull for? expressed as relative but need to say what eg relative to bodyweight? Was there a baseline strength difference between the groups? This has implications for the lack of difference in results.
- Muscle soreness Measures: Is there any evidence to say the passive and active methods of muscle soreness are reliable and valid? Why did you not use pain pressure threshold?
3. Results:
This section was concise and very well presented.
4. Discussion:
overall make a greater discussion of the positive findings ie. CK and Mb results. How do the numbers compare to other studies? are the increases comparable? Would also benefit from greater discussion of why the strength and pain measures showed no between group difference.
Line 10: Minor English - assume you mean overweight not overfat
Line 15-25 put in introduction
Line 26-37: this whole section is not needed. You are going into detail discussing oxidative stress even though this was not part of your data collection and analysis.
Overall:
The CK and Mb results are interesting however I am concerned that the reasons for the lack of significance with respect to pain and strength is due to the methodology employed in measuring these variables.
Reviewer 3 Report
This study addresses the question of the effect of body fat percentage on outcomes of 50 repetitions of eccentric muscle actions in young healthy males.
Major
The abstract is clear and informative.
The Introduction is well written and provides a comprehensive review of the relevant literature.
In the materials and Methods section please provide the ethics registration number for the study.
The results section is clearly written.
The Discussion is clearly written and conclusions drawn from the findings of the study are rational and based on previous research.
Conclusions are reasonable and clearly stated.
Please format the References section correctly.
Minor
Page 1, Line 44. Change "their study was only on women" to "their study included women only"
Page 2, Line 10. Change "compared to" to "compared with"
Although the tables in the Results section are clear, I would prefer to see the data presented in figures however, this is simply my preference and a suggestion to the authors.
Page 6, Line 10. Please delete "overfat." I understand Margaritelis et al (2019) used the term along with "underfat," however, I think is reads better without it.
The limitations you have identified provide sound advice for future research.
Reviewer 4 Report
There is substantial mention of obesity in the manuscript but it is not clear what the body weight category, based on BMI, was for the participants. I suggest to add that information as a subject characteristic.
I suggest to use in the introduction the reference Hickner RC, Mehta PM, Dyck D, Devita P, Houmard JA, Koves T, Byrd P. Relationship between fat-to-fat-free mass ratio and decrements in leg strength after downhill running. J Appl Physiol (1985). 2001 Apr;90(4):1334-41.
CK and Mb were different between the groups but the conclusions at end abstract and conclusion section “may”. Please have the conclusion specific to the findings.
P1, L11. Space between Abstract: and This
P1, Ls 12-13. Change “22.0±1.5” to “22±2” when recorded as whole numbers.
P1, Ls 12-13 Change “176.9±5.2” to “177±5” when measured to nearest cm.
In the abstract, ensure the same font is used.
P1, L22. Change “that may potentially alter muscle” to “that alter the muscle”
P1, L33. Change “exercise, Paschalis” to “exercise. Paschalis”
P1, L33. Clarify whether males or females were participants in the Pachalis et al study.
P1, L35. Change “normal group” to “normal weight group”
P1, Ls 35-36. What is considered high for BMI and BF%. Please provide a number or range what was reported in ref 5 and 7.
P1, L39. Change “animal models and not humans,” to “an animal model and not humans,”
P2, L2. Change “and have a” to “and has a”
P2. L13. Change “22.0±1.5” to “22±2” when recorded as whole numbers.
P2, L13. Change “176.9±5.2” to “177±5” when measured to nearest cm.
Table 1. Age and height in whole numbers when age recorded in whole numbers and height measured to the nearest cm.
P3, L13. “converted to a relative value”. Please clarify.
P3, L26. Muscle soreness is mentioned to be measured immediately after but not presented in Table 3. Please delete “immediately”
P3, L29. Blood was samples immediately after but not presented in Table 5. If you did sample immediately after please present the values for CK and Mb in the tables.
P3, Ls 34-36. Remove bold of “Ortho Johnson Vitros DT60 II” and remove underline of Saluggia
All Tables have as title start “Changes in” but the pre values are not changes. I suggest to delete “Changes in” from the titles of the Tables. The “Changes in” can also be deleted from the headings.
P6, L10. Delete “overfat” and provide the value for BF%.
P6, L11. Replace “escalation” with “elevation” or “increase”
P6, L15. There is discussion of effect of obesity. I suggest to provide the BMI values of your participants groups and whether they were considered obese.
P6, L28. “high %BF”. Please provide the value.
P7, L12. Change” may alter” to “alter”.
P7, L15. “using more diverse parameters.” Please be specific. Which parameters?
The references need to have a consistent style according to author guidelines of the journal.
The references need to have a consistent style according to author guidelines of the journal.
Round 2
Reviewer 1 Report
According to the attached suggestions

Reviewer 2 Report
Dear Authors
I would again like to commend you for the changes already made that I feel have mad a good improvement to the manuscript.
Whilst you have provided a good argument for your use of the eccentric protocol/Isometric strength testing/ Pain measurement methods employed, I do still have some reservations about them in the context that strength and pain in particular did not differ between the groups. This is despite the Ck and Mb levels being significantly different. I therefore pose 2 potential explanations for this.
- Is it that CK and Mb are more sensitive measures of muscle damage, with other measures of strength and pain not being able to differentiate between groups? How does this compare to other studies?
- The assessment methods are not sensitive/specific enough to detect changes? I direct the authors to a paper assessing the correlations between PPT and VAS;
Visual Analog Scale and Pressure Pain Threshold for Delayed Onset Muscle Soreness Assessment
By: Lau, Wing Yin; Muthalib, Makii; Nosaka, Kazunori
JOURNAL OF MUSCULOSKELETAL PAIN Volume: 21 Issue: 4 Pages: 320-326 Published: DEC 2013
The passive measurement of pain in your manuscript involved no palpation pressure as far as is described. This does not make sense to me and is an obvious reason as to why there was no differences found.
Given that the impact of DOMS is an increase in pain and a loss of functional muscle performance, your results suggest that body fat % does not effect these 2 parametres. Therefore some further explanation of the above possibilities for this lack of difference is required in the discussion and limitations section.
I look forward to seeing the revised changes.
Regards
